# Humane Euthanasia of Guinea Pigs (*Cavia porcellus*) with a Penetrating Spring-Loaded Captive Bolt

**DOI:** 10.3390/ani10081356

**Published:** 2020-08-05

**Authors:** Shari Cohen, Melody Kwok, Joel Huang

**Affiliations:** 1Office of Research, Ethics, and Integrity, University of Melbourne, Melbourne 3010, Australia; melodyk@student.unimelb.edu.au; 2Agriculture Victoria, Attwood 3049, Australia; joel.huang@agriculture.vic.gov.au

**Keywords:** guinea pig, Cavia porcellus, captive bolt, humane euthanasia, laboratory animal, humane killing

## Abstract

**Simple Summary:**

Various euthanasia methods are currently employed for guinea pigs at their experimental or humane endpoint; however, many have significant limitations or negative animal welfare implications. Captive bolt euthanasia has been used in the guinea pig meat industry but has not been explored in a research setting. This work aimed to investigate the use of a penetrating spring-loaded captive bolt gun as a refinement to guinea pig euthanasia in research. The study found that when compared to blunt force trauma, the captive bolt procedure performed well against all parameters of humane slaughter of production animals and appears to be a feasible refinement for animal welfare.

**Abstract:**

Guinea pigs (*Cavia porcellus*) have been used in research since the 19th century to collect data due to their physiological similarities to humans. Today, animals perform a vital role in experiments and concerns for laboratory animal welfare are enshrined in the 3R framework of reduction, refinement and replacement. This case study explores a refinement in humane euthanasia of guinea pigs via the use of an irreversible penetrating spring-loaded captive bolt (CB). Penetrating spring-loaded CB stunning for euthanasia (CBE) was performed on 12 guinea pigs with the parameters for humane slaughter of production animals in order to assess the suitability of this method of euthanasia in contrast to blunt force trauma (BFT). All 12 of the guinea pigs were rendered immediately unconscious with excellent experimental tissue quality collection, high repeatability of results and operator (*n* = 8) preference over BFT. Overall, CBE in guinea pigs appears to be a feasible refinement for animal welfare, human preference and improved tissue quality for experimental collection in settings where uncontaminated tissues are required.

## 1. Introduction

For over 200 years, guinea pigs have played an important role as an animal model in biomedical research due to their greater physiological parallels with humans in comparison to other species of rodents and rabbits [1,2]. There is an increasing level of awareness and importance of animal welfare at each stage of research with animals which includes ensuring a humane death. A humane death or euthanasia is defined as achieving immediate stress-free insensibility and death [3,4,5] The very definition of humane euthanasia supports the 3R’s framework of replacement, reduction, refinement as proposed by Russell and Burch in 1959 [6]. The full replacement of animals in some areas of research can be difficult, and at times animal use may still be required. Whenever the use of animals cannot be replaced, it is incumbent on the research community to apply the principle of refinement at all stages of research to improve animal welfare and care [6].

Captive bolt (CB) is a technique that can be applied in the humane euthanasia of animals. Despite its benefits during humane slaughter, it is yet to be applied in the research setting for the collection of uncontaminated non-brain tissue. The CB devices used can be hand-held with a retractable bolt powered by either a spring, compressed air, gun powder or a blank cartridge, and placed on a specific area on the cranium of the animal [7]. Once triggered, it will lead to either stunning (reversible loss of consciousness) or death [7]. CB placements and their effectiveness differs between various animals. [4,7,8,9,10] There are two types of CB: non-penetrating and penetrating [7]. Non-penetrating CB leads to stunning via the transfer of kinetic energy into the cranium and brain and is usually followed by a secondary euthanasia method [4,5,7,11]. In contrast, penetrating CB results in death due to the bolt fracturing the skull and entering the brain, subsequently causing irreversible physical damage to the cerebral cortex, brain stem and the thalamus [10,12,13,14]. The five indicators of successful stunning are lack of corneal blink reflex, rhythmic breathing, righting reflexes, vocalization and head/neck tension [5,15,16]. These indicators are reliable to assess the effectiveness of the shot as CB damages the region of the brain responsible for these responses [12,15,16]. The technique has not been tested in a research setting using a non-commercial captive bolt device and a methodology for laboratory use is yet to be developed. The application of this technique is thought to be transferable and potentially valuable in animal research models requiring chemically uncontaminated and intact non-brain tissues.

The use of a captive bolt to perform humane euthanasia (CBE) has been demonstrated through numerous experiments to be efficacious and humane across a wide range of species, including: guinea pigs [13], rabbits [17], piglets [8,18,19,20,21,22], joeys [11,23], ruminants [24,25,26,27], and avian species [28]. Some of these results showed CBE produced immediate death in 100% of animals [28]. The effective use of this method has also been documented in guinea pigs for humane slaughter in the meat industry, which demonstrates its feasibility in this species [22].

While blunt force trauma (BFT) is a common technique to euthanize guinea pigs in research requiring chemically uncontaminated tissues, there are no known published studies assessing the use of BFT in this species. In other vertebrates, this technique has been compared to CB. In a study based on 170 rabbits by Walsh et al., CBE was shown to be more effective than BFT, with BFT failure rates of up to 23% due to insufficient force [17]. Likewise, studies on piglets, sheep/lambs, and kids have also shown that CBE will produce extensive brain damage before pain can be perceived and caused instantaneous death. In these studies, the failure rate was between 0–6% in animals [8,18,19,22,25,26,27]. This is in contrast to a study on piglets which determined that BFT had a similar failure rate of up to 24%, hence the need for repeated blows [22].

Compassion fatigue is a form of post-traumatic stress disorder. It is defined as exhaustion from the stress of feeling empathetic towards animals and feeling unable to help them in any way [29,30]. This important concept is now readily recognizable in staff working with animals. Compassion fatigue is believed to be exacerbated by performing euthanasia on a regular basis and is partly dependent on the type of method used [29]. A study performed by Rohlf et al. on 148 animal workers in animal shelters, veterinary clinics and research laboratories reported that they experienced mild and moderate stress symptoms (39% and 11%, respectively) from performing euthanasia [31]. Similarly, a systematic review conducted by Scotney et al. demonstrated staff involved in performing euthanasia (animal shelters, veterinary clinics or research laboratories) was correlated with increased work stress and was hence a contributing factor to developing compassion fatigue [32]. In particular, laboratory staff may experience guilt when euthanizing research animals [33]. Therefore, if the euthanasia method has high animal welfare and acceptable aesthetics, then the staff member or “operator” may be less likely to experience compassion or emotional fatigue.

As part of a study requiring the collection of chemically uncontaminated tracheal tissues from guinea pigs (University of Melbourne Animal Ethics ID: 1814500.3), researchers would often use cranial blunt force trauma (BFT) followed by exsanguination or decapitation [34,35]. However, facility staff, animal welfare veterinarians and researchers all expressed concerns with BFT. It was found to be a method operators wished to avoid and has been shown in other studies to potentially have a risk of low repeatability due to inappropriate force or technique, and it is very dependent on the individual operator [5,35,36]. Furthermore, there are known concerns that this method of euthanasia can lead to physical, emotional and compassion fatigue in personnel due to the human–animal bond [5,18,37,38,39,40,41]. Prior to identifying CBE as an alternative technique to BFT, other methods of humane euthanasia were considered. Table 1 outlines the currently available experimental methods of guinea pig euthanasia with their known respective advantages and disadvantages.

After a literature review of possible known options, it was extrapolated that CBE could be suitable for non-brain tissue collection with no chemical residues and possible improvements in experimental, animal and human welfare outcomes when compared to the BFT method [5]. The aim of this case study was to develop and trial a protocol to refine the method of humane euthanasia of guinea pigs for research requiring the collection of uncontaminated non-brain tissues. This paper introduces an alternative application of an existing method of humane euthanasia in a novel setting using CB and the criteria for humane slaughter to ascertain both the practicality and humaneness (animal and human) of using CBE on guinea pigs [15] for research.

## 2. Materials and Methods 

Equipment: A penetrating spring-loaded CB called ‘The Ballista’ (Bunny Rancher, Shapleigh, ME, USA), was used in this research (see Figure 1 and Figure 2 below). This hand-held device was placed mid-forehead on the guinea pigs (see Figure 2 below). The device delivered 6.7 joules with each shot [51].

Parameters to determine humane euthanasia: There are currently no specific recommendations for CB use in guinea pigs. Hence, the livestock parameters of an absence of corneal blink reflex, rhythmic breathing, righting reflexes, vocalization and head/neck tension were utilized for this study.

Experiment: Operators first trialed the device on oranges to gain familiarity with the device. Excess breeding stock of male rat cadavers weighing 250 g or more were used to trial the device prior to use in guinea pigs. Twelve tri-color short-haired American female and male guinea pigs aged 16–22 weeks, weighing 500 g or more, were sourced as excess stock. Guinea pigs were euthanized using CBE. Accurate placement of the CB was first practiced on euthanized rats retrieved from other experiments to decrease the number of guinea pigs needed. All guinea pigs used were part of another animal ethics approved research project (University of Melbourne Animal Ethics ID: 1814500.3) as the researchers agreed to use this method as their humane euthanasia technique. Two personnel were required, with a total of 8 operators performing this method. Prior to the procedure being performed, each guinea pig was moved to a quiet place away from the others. One operator wrapped and held the guinea pig in a towel with its head resting on a rolled-up towel to elevate the head, increase comfort and decrease stress. The second operator pre-loaded the spring prior to positioning the penetrating spring-loaded CB on its head at the intersection between lines drawn from the base of the ears with the contralateral eyes (see Figure 2). When triggered, a retractable bolt was fired with the aim of producing effective and instantaneous unconsciousness. Animals were unwrapped to assess the 5 livestock humane euthanasia parameters. This was determined using five indicators (absence of corneal blink, rhythmic breathing, audible vocalization, righting reflex and head/neck tension) in each individual. Exsanguination was performed within the next 20–30 s. The results were recorded in and reviewed.

## 3. Results

All 12 guinea pigs failed to show signs of corneal blink, righting reflex, audible vocalization, head/neck tension and rhythmic breathing immediately after the initial shot. Based on these criteria, all guinea pigs were considered to have been stunned and rendered unconscious immediately. No additional shots were required for any of the guinea pigs. Exsanguination via severance of the lower abdominal aortic vessel was performed within 20–30 s of the initial shot to ensure death. A few of the guinea pigs were seen to display signs of involuntary muscle fasciculations and slow hindlimb pedaling motions, which lasted for less than 30 s. The bolt produced cranial skull fractures with associated subcutaneous hemorrhages, but the overlaying skin remained intact. While there was no evidence of puncture or bleeding overlying the site where the bolt penetrated the skull (Figure 3 and Figure 4), there was a discrete palpable, circular depression in the cranium the size of the captive bolt diameter. Mild epistaxis (*n* = 3) was seen post-CBE. Informal debriefing with the operators (*n* = 8) and researchers (*n* = 4) suggested they all favored CBE over the BFT method. They stated CBE had a greater yield of higher quality uncontaminated tissue and was less physically, emotionally and compassionately fatiguing than BFT [52].

## 4. Discussion

This study demonstrated that the penetrating spring-loaded CBE device can be a useful technique in the humane euthanasia of experimental animals by successfully producing instant and irreversible unconsciousness in 100% of the guinea pigs. This assessment of loss of consciousness is based on the absence of reflexes and the significant palpable physical damage to the skull, indicating sufficient damage to the vital regions of the brain [5,16,53]. The small but consistent result in this study exceeds the accepted animal welfare standard proposed by Temple Grandin for livestock in abattoirs, whereby 95% of animals should be immediately stunned with a single shot [54]. It also further supports the results from the study conducted by Limon et al. on South American commercial guinea pigs and is comparable to findings for other vertebrate species. In this study, some guinea pigs displayed involuntary muscle fasciculations and slow hindlimb pedaling motions after the initial shot. Both pedaling and head/neck movements are known autonomic movements controlled by reflex circuits in the spinal cord; thus, they commonly occur even after spinal severance due to the residual activity of these circuits [7,53,55].

The CB device used in this study was purchased from a commercial supplier which advertised it as a penetrating CB device. From the above results, all the skulls had extensive fractures, which is characteristic of a penetrating CB, although the skin remained intact after the procedure. This appearance is more characteristic of a non-penetrating CB which has a concussive effect without the bolt entering the cranium or penetrating the skin. Hence, while the skin remained intact, this paper refers to the device as a penetrating CB based on the photographs post-euthanasia as the bolt has clearly entered the cranium. An unexpected outcome of the experiment was that the overlying cranial skin of all guinea pigs was not compromised and remained intact. This observation was different to what was anticipated, especially given the extent of the fractures observed, and is not a typical characteristic of a penetrating CB. The intact skin may potentially be the result of the increased elasticity of the skin around the head in a guinea pig compared to the skin of other animals, such as cattle and rabbits, where the skin is more tightly adhered to their skulls. The cranial fractures associated with the position of the CB on the skull was expected given the nature of the device. The combination of the absence of measured parameters and post-mortem examination (hemorrhages and fractures) demonstrated that CBE resulted in effective stunning and insensibility in all the guinea pigs studied, even though the overlying skin was not compromised.

### 4.1. Effectiveness of Captive Bolt in Animals

Currently, there is only one published review of CBE in guinea pigs. This study was performed on South American commercial guinea pigs and four different slaughter methods were compared (cervical dislocation, electrical head-only stunning, carbon dioxide stunning and penetrating CB) [13]. Similarly to this paper, a small sample size of guinea pigs was used (10) with one guinea pig requiring a second shot. This could be argued as a 10% fail rate which is above the Temple Grandin recommendations for humane euthanasia with CB. Possible reasons for this guinea pig requiring a second shot may have been related to handling and restraint, incorrect placement of the CB, differences in devices or device failure. These problems have been described in other studies investigating CB in other species, in which insufficient pressure and/or penetration of the device, inaccurate placement of the device, inappropriate CB chosen for the intended species or prior head injury in animals were associated with failure of CBE or the need for multiple attempts [10,20,23,26,28]. These are all aspects to be managed and explored appropriately when using CBE. It is noteworthy that the use of CBE in our study for experimental tissue collection did not result in failure for any animal involved, despite the relative inexperience of the researchers with this technique.

The findings from these papers, along with this present study, support the conclusion that CBE can be an effective and humane method of euthanasia across many vertebrate species. In contrast, BFT has been shown to have a comparatively high failure rate. BFT also has the potential to more frequently fail the international standard of having at least 95% of animals being stunned after an initial blow/strike to preserve animal welfare. Therefore, when compared to CBE, BFT has a higher potential to negatively impact the welfare of guinea pigs and other vertebrate animals. The use of CBE appears to support the 3R framework principle of refinement by utilizing a more humane and preferable method to BFT in animals used for research.

### 4.2. Uncontaminated Tissues

At times, high quality chemically uncontaminated non-brain tissues are required for medical or biomedical research purposes. According to the operators in this study, the tracheal tissues collected were of a higher quality compared with tissues obtained from previous experimental work utilizing the BFT method. In these types of experiments, the majority of other euthanasia options available for guinea pigs (see Table 1) are not ideal as they have the potential to contaminate tissues, decrease tissue quality and affect metabolic serum biomarkers [45,46]. These issues can be potential confounders that can affect or compromise experimental outcomes, which could mean the information obtained may be incorrect or inconclusive. As such, the use of CBE over BFT supports the 3R principle of reduction as fewer animals may be required when higher quality tissues are used with better results. By utilizing high quality tissues, fewer animals may be used as research outcomes which may be more robust with a lower rate of rejected tissues. In regards to guinea pigs raised for human consumption, there is a legal and ethical requirement to ensure that meat products do not have any chemical residues unsuitable for human consumption and that animals are humanely killed. The use of CBE can be seen as a preferred method to support refinement due to potential improvements in animal welfare and research outcomes whenever intact brain or nasal tissues are not required [5].

### 4.3. Physical, Emotional and Compassion Fatigue

Operators anecdotally reported increased satisfaction with using a CBE compared to BFT due to the decreased variability of the effectiveness and easier application of the technique. This could potentially reduce the likelihood of physical, emotional and compassion fatigue being experienced by the operators. This is similar to statements made by other operators in different experiments which also rated CBE as the most aesthetically pleasing method of euthanasia when compared to BFT [8,17,22]. Aspects of operator fatigue include physical, emotional and compassion fatigue. Physical fatigue can be reduced when the CB is correctly placed on the cranium as the kinetic energy is delivered by the device [56,57]. This is different to the use of BFT which relies on the force applied by operators.

Emotional and compassion fatigue can also be linked to the human–animal bond. This bond is thought to be a contributing factor in the compassion fatigue experienced by people working in the animal profession [29]. Thus, by reducing the strain on the human–animal bond, both emotional and compassion fatigue can possibly be reduced. In this study, the operators using the CB reported this method to be more aesthetically appealing, as the bolt did not appear to significantly compromise the skin (Figure 3 and Figure 4). They also felt the method of euthanasia was more humane. As a result, this method appeared to be less confronting and overall more appealing to operators, and potentially less detrimental to their wellbeing.

While the human element is an important aspect of humane euthanasia, the psychological aspect of this euthanasia technique was not objectively assessed in this paper. However, the initial anecdotal evidence appears to be positive. In future studies, a survey could be used to assess the emotional effect of CBE use to further investigate the human welfare aspect of various techniques of humane euthanasia. Overall, there was a verbalized positive impact on the physical and psychological well-being of operators when using this method in this case study. However, for CBE to be successful in future experimental studies or settings, adequate training and correct restraint of the animals must be ensured, and an appropriately sized CB device must be used and maintained. Other causes for CBE failure can arise if personnel are not adequately trained, which can lead to incorrect restraint of animals and placement of the device. If any of the above are performed incorrectly, higher levels of physical and emotional fatigue can occur due to failure of the device or technique [5,7,8,9,20]. Therefore, correct training and device usage is essential to decrease the chances for physical, emotional and compassion fatigue, as well as reducing risks to personnel and animals [3,36,58].

### 4.4. Limitations and the Future

Due to the small sample size of guinea pigs, aspects such as device fatigue and physical fatigue experienced by operators from loading the spring with each guinea pig were not evaluated. Other potential parameters that determine insensibility and success of stunning include measuring auditory evoked potential (AEP) and electroencephalogram (EEG) activity with electrodes. Both correlate to the cessation of convulsions, loss of reflexes and brain death [8,59,60]. These were not used in this study as the five parameters for humane livestock slaughter were deemed adequate to indicate insensibility. The use of EEG in CBE also risks damaging the electrodes and was deemed inappropriate for this study [8].

Additional parameters to measure could include the absence or presence of a heartbeat. However, cardiac contractions can continue in unconscious or brain-dead animals and thus this parameter was not used [8,18,61]. Whilst histological changes or scoring of the damage to the cranium and brain could have been assessed (skull fractures were easily observed and palpated), the degree of brain and cranial damage was not considered essential in this study [8,13,17,18]. An area that would benefit from additional investigation would be to explore alternate types of restraint and equipment. The current technique described requires a handler and a CB operator. It would be beneficial to develop or modify a device to enable the task to be performed by only one CB operator.

Finally, a different type of CB device could be tested. The use of an industrial CB (gunpowder-powered, compressed air or CO_2_ blank cartridges) may be more suitable than the spring-powered CB in larger cohorts or in an abattoir setting for animals bred for human consumption. This is because the spring-loaded CB is designed to be used for <1000 animals per year and can experience device fatigue more quickly compared to an industrial CB [51]. The downside of an industrial CB is that they are more expensive and require more comprehensive training to use. However, the use of an industrial CB can result in less physical fatigue compared to a spring-loaded CB as there is no requirement to physically pull the spring with each use. Thus, the use of industrial CBs could be considered if a greater number of animals are to be used in a study due to larger cohorts or in other settings (i.e., an abattoir). It should also be noted that as of 2020, within Australia, CB is an accessible humane euthanasia technique as there are no legal requirements to hold a firearms license when using a CB in six out of the seven Australian states and territories. The authors are aware of similar legislation in other countries, which makes CBE a potentially accessible and humane euthanasia technique internationally in research and other settings.

From this study and other studies that have been undertaken, it can be tentatively proposed that CBE can be used as a suitable alternative for the collection of non-brain uncontaminated tissues. A drawback of this study is the small sample size available at the time. Further research should be undertaken to review the potential failure rate on a larger population to determine potential complications. Future studies in refinement could explore different handling/restraint techniques or devices, and the effects of the use of different CBs. Additional work can be performed to ensure consistency of the results from this study. As compassion and emotional fatigue are so prevalent, any future studies should incorporate formal surveys of operator preferences regarding their levels of physical, emotional and compassion fatigue and how this changes when performing euthanasia in different conditions.

## 5. Conclusions

In conclusion, this small study has shown CBE can be an ideal and humane method compared to other techniques (i.e., BFT) when uncontaminated tissues are needed. Thus far, it has been shown to have favorable experimental, animal and human welfare outcomes in guinea pigs where high quality, uncontaminated non-brain tissue collection is required. Future research could increase the sample size and use alternative parameters (e.g., AEP, EEG, post-mortem and histological changes) to corroborate the evidence for the effectiveness of CBE. Other areas to be explored could be the use of an industrial CB, alternative restraining methods or devices, and the inclusion of formal surveys to evaluate operator attitudes and experiences in performing CBE.

## Figures and Tables

**Figure 1 animals-10-01356-f001:**
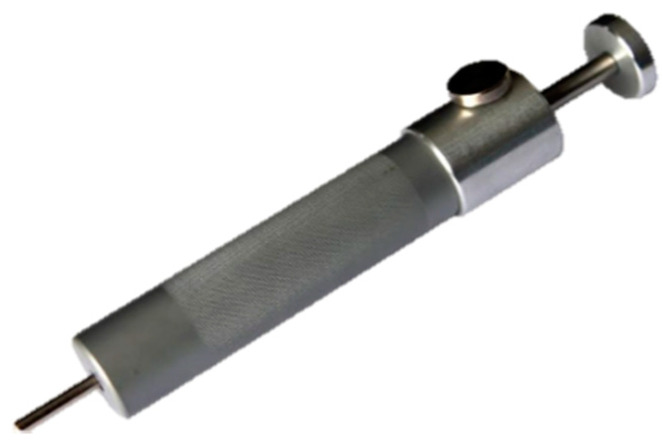
‘The Ballista’: a penetrative spring-powered captive bolt (CB) gun used for the euthanasia of guinea pigs in this research. Image used with permission [51]

**Figure 2 animals-10-01356-f002:**
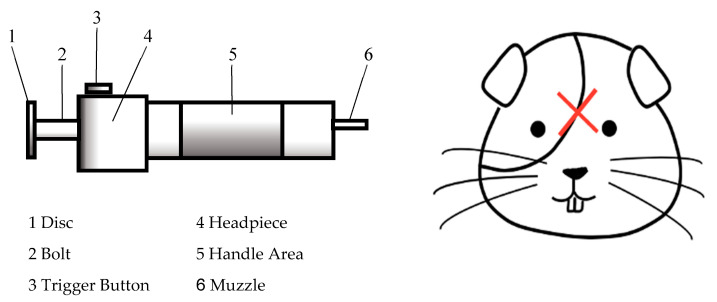
(**Left**): ‘*The Ballista*’ spring-powered non-penetrating captive bolt. (**Right**): The red cross denotes where the captive bolt was positioned on the guinea pigs.

**Figure 3 animals-10-01356-f003:**
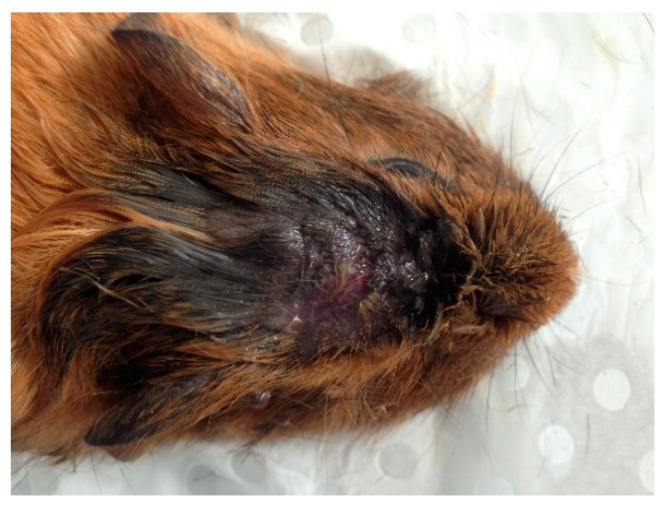
Skin intact from penetrative CB site.

**Figure 4 animals-10-01356-f004:**
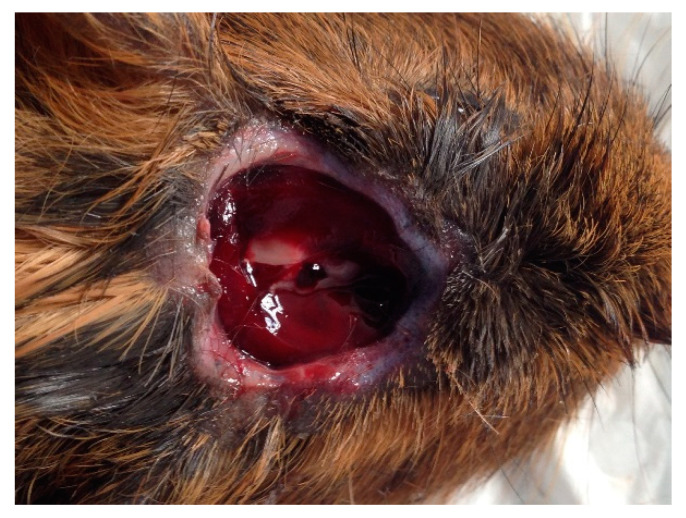
Skin removed from penetrative CB site.

**Table 1 animals-10-01356-t001:** Advantages and disadvantages of alternative euthanasia methods [5,16,34,35,36,42,43,44].

Euthanasia Method	Disadvantages	Advantages
**Anesthetic Gas Overdose**	● Time consuming (slow accumulation in the lungs) [5,16,35,43]. ● Special equipment needed [5,35,42,44]. ● Exposure to agents may be harmful to operators [5,35,42,44]. ● Contaminated tissue [45]. ● Restricted substance (Schedule 4) [35].	● Useful in small animals or when veins are inaccessible [5,35]. ● Different methods available e.g., face mask, containers, etc. [5]. ● Non-flammable, non-explosive gases [5]. ● Both a stunning and euthanasia method [5].
**Barbiturates**	● Increased alkalinity and concentration can cause pain if injected intraperitoneal [34,35,45]. ● Intra-cardiac only in unconscious animals [34,35,42]. ● Contaminated tissue [5,34,45]. ● Tissue artefact [5,34,45]. ● Intravenous ideal but greater training required [5,34,36]. ● Restricted substance (Schedule 4) [35].	● High speed of action (dose, concentration, route, rate dependent) [5,34]. ● Inexpensive [5]. ● Smooth euthanasia pending route [5].
**Carbon Dioxide Stunning**	● Time consuming, especially in immature animals [5,8,16,44,45]. ● Onset of unconsciousness not always immediate [5,43]. ● High CO_2_ concentrations can lead to distress (vocalization, dyspnea) and/or mucosal pain and post-mortem pulmonary and upper respiratory tract lesions [16,34,35,43,45,46,47,48,49,50]. ● Guinea pigs must be <600 g in Victoria, Australia [42]. ● Harmful to operators if significant amounts inhaled [35,43].	● Not a restricted substance and readily available in cylinders [5,35]. ● Inexpensive, non-flammable, non-explosive [5,34,35,43]. ● Minimal tissue contamination [5]. ● Can potentially induce rapid unconsciousness [5,34].
**Cervical Dislocation**	● Aesthetically displeasing [5,34,45]. ● Guinea pigs must be <150 g in Victoria, Australia [42]. ● Extensive training needed [5,34,35,42]. ● Challenging due to anatomical differences to rats/mice (shorter neck with more muscle covering) [36,45]. ● Electrical activity may persist for 13 s after the procedure, which may indicate consciousness and pain perception [5,16,35]. ● Force dependent [17]. ● Damage to brain tissues [5].	● Inexpensive. ● Can result in rapid loss of sensibility [5,16,34,36]. ● No tissue contamination [5,16].
**Decapitation**	● Aesthetically displeasing [5,16,45]. ● Human safety risk and special training needed [16,36,43]. ● Special equipment is needed (guillotine) [16,36,43]. ● Challenging due to anatomical differences to rats/mice (shorter neck with more muscle covering) [45].	● Inexpensive. ● Can result in rapid loss of sensibility [5,36]. ● No tissue contamination [5,43]. ● Quick method [34].
**Blunt Force Trauma**	● Aesthetically displeasing [5,42,45]. ● Force and accuracy of blow dependent [5,8,35]. ● Issue with repeatability due to fatigue [5,8]. ● Damage to brain tissues [5].	● Inexpensive. ● Effective when applied accurately and with sufficient force [5,35,36]. ● No tissue contamination [42]. ● When accurate with sufficient force, can be a quick method [34,35].

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
