# Peer review of "Humane Euthanasia of Guinea Pigs (Cavia porcellus) with a Penetrating Spring-Loaded Captive Bolt"

_animals, 2020, doi:10.3390/ani10081356_

Round 1

Reviewer 1 Report

Overall, some of the syntax used in the manuscript can be difficult to follow.  For example, the second sentence in the introduction is quite awkward to read.

The introduction would be strengthened with a description of when BFT or CBE might be preferable to anesthetic overdose or other methods of euthanasia.  Although this information is eventually provided, explaining the need to avoid chemical contamination early in the introduction would help the reader understand why the study was performed.

Line 56:  If this method has been demonstrated to be effective in the meat industry, how does your study build on this and provide new information?  Is it because of the penetrating versus non-penetrating?

Lines 116-118 and 131-133:  Please clarify that you are looking for an absence of these signs.  This is not clear as written.

Lines 119-121:  How old were the guinea pigs?  What sex were the guinea pigs?  What else had been done to them?  What was the strain of guinea pig that was used?

Figure 2:  The labels for 3 and 6 are not visible.

Author Response

Overall, some of the syntax used in the manuscript can be difficult to follow.  For example, the second sentence in the introduction is quite awkward to read.

Revised.

The introduction would be strengthened with a description of when BFT or CBE might be preferable to anesthetic overdose or other methods of euthanasia.  Although this information is eventually provided, explaining the need to avoid chemical contamination early in the introduction would help the reader understand why the study was performed.

Amended.

Line 56:  If this method has been demonstrated to be effective in the meat industry, how does your study build on this and provide new information?  Is it because of the penetrating versus non-penetrating?

Included.

Lines 116-118 and 131-133:  Please clarify that you are looking for an absence of these signs.  This is not clear as written.

Amended.

Lines 119-121:  How old were the guinea pigs?  What sex were the guinea pigs?  What else had been done to them?  What was the strain of guinea pig that was used?

Included.

Figure 2:  The labels for 3 and 6 are not visible.

Amended.

Reviewer 2 Report

Dear Authors,

I have read your manuscript with interest and take this opportunity to make suggestions. 

General:

  • Abbreviations: as a rule explain abbreviations the first time they appear in the text. Two examples: Introduction, second paragraph: 'Captive bolt is a technique ...' change into 'Captive bolt (CB) is a technique ...'. Introduction, fourth paragraph: 'However, overall CBE has been ...', CBE is not explained. 

Materials and Methods: 

  • Information on the individual animals is not presented: sex, age/ weight/ size?
  • Animals were wrapped in a towel: how were the rhythmic breathing, righting reflex and head/neck tension determined immediately after firing the bolt?
  • I assume that with 'vocalisations' are meant only those audible for the human ear?!

Results:

  • Figure 3 and 4: would it be possible to still add ruler measurements? Without indication of relative size, it is difficult to assess the extent of the trauma caused, although it is stated in the text that it is of similar size as the diameter of the bolt.  
  • Exsanguination: could you provide indication of the ratio total volume collected and the total volume of blood per animal? This ratio may indicate the extent of continuing heartbeat. 

Discussion:

  • Check text sentence 202-203: 'Therefore, when compared to CBE, has a higher potential to negatively impact the welfare of guinea pigs and other vertebrate animals'. Add 'BFT'?
  • Check text sentence 210-212: 'As such, the use of CBE over BFT also supports the 3R’s principle of reduction as fewer animals are required from these higher quality tissues and better results.' It reads as if something is missing between 'and' and 'better'.

Author Response

General:

  • Abbreviations: as a rule explain abbreviations the first time they appear in the text. Two examples: Introduction, second paragraph: 'Captive bolt is a technique ...' change into 'Captive bolt (CB) is a technique ...'. Introduction, fourth paragraph: 'However, overall CBE has been ...', CBE is not explained. 

Revised.

Materials and Methods: 

  • Information on the individual animals is not presented: sex, age/ weight/ size?
  • Animals were wrapped in a towel: how were the rhythmic breathing, righting reflex and head/neck tension determined immediately after firing the bolt?
  • I assume that with 'vocalisations' are meant only those audible for the human ear?!

Amended and revised.

Results:

  • Figure 3 and 4: would it be possible to still add ruler measurements? Without indication of relative size, it is difficult to assess the extent of the trauma caused, although it is stated in the text that it is of similar size as the diameter of the bolt.  

Figure 3 & 4. Unfortunately we would be unable to accurately apply a ruler as we did not measure the size of the hole or the head. We will take that into consideration for future studies and appreciate this comment. 

  • Exsanguination: could you provide indication of the ratio total volume collected and the total volume of blood per animal? This ratio may indicate the extent of continuing heartbeat. 

The primary purpose of this study was not the collection of blood or the use of CB but the immediate removal of upper respiratory tract tissues for primary research. The tissues collected must be used within approximately 5 minutes of euthanasia for the primary experimental aims to be successful. Therefore the blood was not collected. We will take that into consideration for future studies whenever possible and appreciate this thoughtful comment. 

Discussion:

  • Check text sentence 202-203: 'Therefore, when compared to CBE, has a higher potential to negatively impact the welfare of guinea pigs and other vertebrate animals'. Add 'BFT'?
  • Check text sentence 210-212: 'As such, the use of CBE over BFT also supports the 3R’s principle of reduction as fewer animals are required from these higher quality tissues and betterresults.' It reads as if something is missing between 'and' and 'better'.

Amended and revised.